# PVP/Highly Dispersed AgNPs Nanofibers Using Ultrasonic-Assisted Electrospinning

**DOI:** 10.3390/polym14030599

**Published:** 2022-02-02

**Authors:** Li Zhu, Wanying Zhu, Xin Hu, Yingying Lin, Siti Machmudah, Hideki Kanda, Motonobu Goto

**Affiliations:** 1Department of Materials Process Engineering, Nagoya University, Furo-cho, Chikusa-ku, Nagoya 464-8603, Japan; zhu.li@g.mbox.nagoya-u.ac.jp (L.Z.); zhu.wanying@k.mbox.nagoya-u.ac.jp (W.Z.); hu.xin@i.mbox.nagoya-u.ac.jp (X.H.); lin.yinging@g.mbox.nagoya-u.ac.jp (Y.L.); wahyudiono@b.mbox.nagoya-u.ac.jp (W.); 2Department of Chemical Engineering, Institut Teknologi Sepuluh Nopember, Surabaya 60111, Indonesia; machmudah@chem-eng.its.ac.id

**Keywords:** atmospheric pressure pulsed discharge plasma, ultrasonic-assisted electrospinning, silver nanoparticles

## Abstract

Silver nanoparticles (AgNPs) are novel materials with antibacterial, antifungal, and antiviral activities over a wide range. This study aimed to prepare polyvinylpyrrolidone (PVP) electrospinning composites with uniformly distributed AgNPs. In this study, starch-capped ~2 nm primary AgNPs were first synthesized using Atmospheric pressure Pulsed Discharge Plasma (APDP) at AC 10 kV and 10 kHz. Then, 0.6 wt.% AgNPs were mixed into a 10 wt.% PVP ethanol-based polymer solution and coiled through an Ultrasonic-assisted Electrospinning device (US-ES) with a 50 W and 50 kHz ultrasonic generator. At 12 kV and a distance of 10 cm, this work successfully fabricated AgNPs-PVP electrospun fibers. The electrospun products were characterized using Scanning Electron Microscopy (SEM), Transmission Electron Microscopy (TEM), High-Resolution TEM (HR-TEM), Fourier Transform Infrared Spectroscopy (FT-IR), X-ray Diffraction (XRD), Thermogravimetric (TG), and X-ray Photoelectron Spectroscopy (XPS) methods.

## 1. Introduction

Silver nanoparticles (AgNPs) with a size of 1–2 nm, due to their unique physicochemical properties, which differ from the metal Ag phases, can be used in a variety of novel applications, including antimicrobial materials, medical device coatings, optical sensors, cosmetics, electronic composites, food packaging, diagnostics, orthopedics, drug delivery, and anticancer agents [1,2,3]. At present, the methods for synthesizing AgNPs include the photochemical method [4], laser ablation method [5], and chemical reduction [6]. However, most of these require different organic solvents, which bring the risk of chemical reactions and biological hazards, or that the production rate of the AgNPs is not high. In this work, we synthesized monodisperse AgNPs using atmospheric pressure pulse discharge (APDP), as it provides an efficient continuous flow reaction for the real-time synthesis of nanoparticles. No organic solvent was used in the whole APDP synthesis process. Moreover, APDP is a fully controllable reaction, and low-temperature synthesis prevents chemical functional group changes in stabilizers [7,8,9]. The AgNPs were fabricated in a slug flow reactor system equipped with a non-equilibrium atmospheric-pressure plasma. Ag ions were transformed into nanometer-scale AgNPs by applying a high alternating current (AC) voltage on the surface of a glass capillary containing aqueous droplets and bubbles. Here, starch has been used as a green capping agent for AgNPs [10]. Due to stabilizer starch, the freshly synthesized Ag nanoparticles did not aggregate to form clusters or blocks.

It is desirable to attach nanoparticles to composite materials, which can be applied in applications including drug encapsulation and controlled release [11,12,13], food preservation [14], wastewater treatment [15], gas sensing [16], and environmental remediation [17,18]. In this work, as one of the composite materials, nanofiber was selected due to its high surface area to volume ratio, efficient mechanical properties, and specific biochemical properties [19,20,21,22]. Here, the electrospinning technique is an efficient method of fabricating nanofibers [23,24]. The attached AgNPs electrospun nanofiber mats can be used in antibacterial wound dressings [25,26,27], air filtration [28], food packaging [29], catalysts [30,31] and conductive films [32]. Here, for increasing the mechanical and biological qualities of fiber products, a highly dispersed nanoparticle is required [33].

However, inorganic or insoluble nanoparticles are not miscible with polymer solutions, resulting in aggregated clusters and uneven striped dispersion of nanoparticles on electrospun fibers. Mechanical stirring for enhancing dispersion uniformity [34] cannot prevent particles from aggregating [35]. It is well known that ultrasonic is a commonly used method for the deagglomeration and dispersion of nanoparticles into liquids [36,37,38,39]. Some studies added surfactants [40,41] and stabilizers [42] simultaneously as ultrasonic treatment. Additionally, existing studies demonstrate that ultrasound can effectively disperse inorganic nanoparticles in polymer solutions. For example, ultrasound effectively disperses Al_2_O_3_ and ZnO nanoparticles in P(VDF-TrFE) [43] as well as Ag [44], Al_2_O_3_ [45], TiO_2_ [46,47], and SiO_2_ [48] nanoparticles in epoxy resin.

We developed a real-time ultrasonic-assisted electrospinning (US-ES) process to achieve uniform nanoparticle dispersion. Ultrasonic waves trigger nanoparticles to absorb destructive energy. The propagation of ultrasonic waves in a liquid medium generates high pressure (up to thousands of atmospheric pressures), destroying the binding energy between particles. The bubbles from cavities invade the new particle gaps to break aggregated particles, forming primary-sized particles [35,49,50]. In addition, a large amount of energy is released from the bubble collapse and transferred into the liquid solution, resulting in a mechanical stirring effect to uniformly disperse the nanoparticles. We used insulated and flexible polyether ether ketone (PEEK) as a pre-nozzle to prevent high-voltage (12 kV) short circuits in an ultrasonic generator. The pipe was tightly coiled around an ultrasonic generator.

In this work, Polyvinylpyrrolidone (PVP) was selected as the host of electrospun polymers. It can be effectively used in medicine, pharmacy, cosmetics, and composite material applications due to its non-toxic nature, biocompatibility, temperature resistance, pH stability, and chemical inertness [51,52,53]. In addition, PVP molecules provide stability to the AgNPs-PVP bond through steric repulsion, which prevents the subsequent aggregation of AgNPs [54]. AgNP-loaded PVP fibers could be applied as glucose sensors [55], and biocompatible materials include antimicrobial mats [56] and wound dressing materials [57]. This study finally fabricated the PVP electrospun fibers with highly dispersed AgNPs using APDP and US-ES without adding surfactants. The characteristics of the electrospun products were also investigated.

## 2. Materials and Methods

### 2.1. Materials

Silver nitrate (AgNO_3_, Product No. 900-00935) and starch (Product no. 191-03985) were purchased from Wako Pure Chemical Industries (Osaka, Japan). Argon (Ar, purity > 99.99%) was purchased from Sogo Kariya Sanso (Nagoya, Japan). Polyvinylpyrrolidone (PVP, ~MW 1,300,000) was purchased from Sigma-Aldrich (St. Louis, MO, USA). All the chemicals used in this study were used as received without further purification.

### 2.2. Methods

#### 2.2.1. Synthesizing of AgNPs Using APDP

Figure 1a shows the synthesis of AgNPs using pulsed discharge plasma in an atmospheric slug flow reactor. This self-made equipment consisted of a capillary glass tube (2.0 mm i.d), a liquid pump (HPLC, LC-10AD, Shimadzu Co., Kyoto, Japan), a gas flow meter (RK-1250, Kofloc Instruments Inc., Kyoto, Japan), and an AC power supply (TE-HVP1510K300-NP, Tamaoki Electronics Co., Ltd., Kawaguchi, Japan). The feed solution was prepared by mixing 6 mmol/L AgNO_3_ and distilled water containing 0.1 wt.% starch. Moreover, it was stored in a desiccator at room temperature. The feed solution and Ar gas were introduced into the upper section of the capillary glass tube simultaneously to react with gas-liquid motion in this slug flow reactor system. The flow rate for the feed solution was 1.5 mL min^−1^, and for Ar gas it was 0.15 mL min^−1^. The HPLC pump was used to control the follow-up rate with the help of gas flow meters and a metering valve near the bottom of the capillary glass tube. After the flow bubbles in the glass capillaries stabilized, a discharge was applied to the slug flow reactor system to generate plasma using a power supply with a bipolar pulsed output voltage. The power and frequency of the AC power supply were maintained at 10 kV and 10 kHz, respectively, during the entire operation. The temperature of the capillary glass surface was maintained under 40 °C and measured using a compact thermal imaging camera (C3-X, FLIR, Wilsonville, OR, USA). The yellow AgNPs solution was vacuum freeze-dried (EYELA FDU-1200, Rikakikai Co., Ltd., Tokyo, Japan) to obtain starch-capped AgNPs powder. Figure 1b shows that the breakdown voltage and current were monitored and determined using a digital oscilloscope (TDS2024C, Tektronix Inc., Yamato, Japan) equipped with a high-voltage probe and a current transformer.

#### 2.2.2. Fabrication of AgNPs-PVP Nanofibers by Ultrasonic-Assisted Electrospinning (US-ES)

The freeze-dried starch-capped AgNPs powder (0.2, 0.4, and 0.6 wt.%), PVP (10 wt.%), and ethanol solvent were mixed by magnetic stirring at room temperature for 2 h to obtain an AgNPs-PVP feed solution. As shown in Figure 2, the polymer solution was injected into a needle through a high-pressure syringe pump (PHD-Ultra 4400, Harvard Apparatus, Holliston, MA, USA) at 0.05 mL/min flow rate in the US-ES process. A 12 kV DC power supply (HARb-30P1, Matsusada Precision, Osaka, Japan) was used to generate a jet at 10 cm from the needle to the collector. The 50 W ultrasonic generator continuously generated 50 kHz ultrasonic waves during the entire electrospinning process. The environment of the ultrasonic process was maintained below 40 °C and measured using a compact thermal imaging camera (C3-X, FLIR, Wilsonville, OR, USA). This setup used a 1 m insulated PEEK tube with an inner diameter of 0.25 mm as a pre-needle tightly coiled around the ultrasonic generator. The electrospinning operations were performed under environmental conditions of 22 ± 1 °C temperature and 25 ± 3% relative humidity after the ultrasonic treatment. The AgNPs-PVP electrospun fiber mats were collected from the fiber collector.

### 2.3. Characterization

The starch-capped AgNPs and electrospun fiber products were characterized using Transmission Electron Microscopy (TEM; JEM-2100Plus, JEOL, Tokyo, Japan), High-Resolution TEM (HRTEM; JEM-2100F/HK, JEOL, Tokyo, Japan), Energy-Dispersive X-ray Spectroscopy (EDS; JED-2300T & Gatan, GIF Quantum ER, JEOL, Tokyo, Japan), Scanning Electron Microscopy (SEM; S-4300, Hitachi, Tokyo, Japan), Ultraviolet-Visible Spectrophotometry (UV-Vis; V-550, JASCO, Tokyo, Japan), X-ray Photoelectron Spectroscopy (XPS; ESCA-3300, Shimadzu, Kyoto, Japan), Thermogravimetric Differential Thermal Analysis (TG-DTA; DTG-60AH, Shimadzu, Kyoto, Japan), and Fourier Transform Infrared Spectroscopy (FT-IR; PerkinElmer Ltd., Waltham, MA, USA).

## 3. Results and Discussion

### 3.1. Synthesizing of AgNPs

#### 3.1.1. Analysis of AgNPs

As shown in Figure 3a, Ag nanoclusters of approximately ~500 nm exhibited a rough surface and non-regular shape in the absence of a starch stabilizer. This is because instantaneous uniform nucleation occurred in the plasma reactor. The nucleation and growth of the nanoparticles in the solution are not sufficiently separated without a stabilizer, leading to the incorporation of monomers into the stable core, resulting in uneven nanoparticles. As shown in Figure 3b, when 0.1 wt.% starch stabilizer was used in the slug flow reactor system, the AgNPs were highly dispersed and the silver nanoparticles (~2 nm) did not aggregate to form clusters or clumps.

As shown in Figure 4a, a yellow color product solution was generated, guiding the formation of AgNPs, compared to the colorless starting 6 mmol/L AgNO_3_ containing 0.1 wt.% stabilizer starch as the raw material solution [58]. Moreover, the Tyndall effect was observed in the AgNPs solution when irradiated by a green laser beam, demonstrating highly dispersed colloidal particles in the AgNPs solution [59]. A characteristic absorption range starting from 400 nm was observed in the UV-Vis spectroscopy analysis of the AgNPs solution. This peck absorption is a unique feature of AgNPs [8].

As shown in Figure 4b, most of the primary particles of AgNPs exhibit a size of approximately 2 nm and a spherical shape with a smooth surface from the TEM image. Nevertheless, several agglomerates on the TEM support films were still found. Probably, this came from the aggregation of starch-capped primary-sized particles to form secondary-sized particles as the AgNPs solution droplets dried. We calculated the mean average size (2.17 nm) and standard deviation (SD = 1.83) of the particles by measuring the sizes of 300 randomly selected particles using Gatan Microscopy Suite (GMS) software.

Figure 4c of the EDS mapping and spectrum shows that the silver element was detected in the starch-capped AgNPs product. The lattice line, measured to study the crystalline properties of the starch-capped AgNPs shown in Figure 4d using the GMS software, was equal to 0.23 nm. Hence, a well-separated single lattice fringe of 0.235 nm corresponds to pure silver particles [60]. As shown in Figure 4e, the agglomerated AgNP lattice lines were at different angles, demonstrating that the large secondary particles (approximately 20 nm in size) were loosely composed of several primary particles (approximately 2 nm in size). From the SEM image, we observed that the freeze-dried starch-capped AgNPs were relatively loose and could decompose under ultrasound. Therefore, ultrasound could effectively decompose and disperse AgNPs into the PVP solution.

#### 3.1.2. Synthesis Mechanism of AgNPs

Figure 5 shows the mechanism of synthesizing starch-capped AgNPs using APDP in a slug flow system. When the gas/liquid is discharged in a pulsed plasma, chemically active substances with reducing properties are generated [9]. High-voltage electrons ionize argon molecules in a local high electric field to form argon radicals. The argon radicals come in contact with the water surface in the slug flow system, resulting in the decomposition of water molecules into hydrogen (H^•^) and hydroxyl radicals (OH^•^) in the solution [61].
(1)$$\mathrm{Ar}\;\overset{\mathrm{plasma}}{\to }{\;e}_{\mathrm{aq}}^{-},{\;\mathrm{Ar}}^{•},\;\mathrm{etc}.$$
(2)$$H_{2}O\;\overset{\mathrm{plasma}}{\to }{\;e}_{\mathrm{aq}}^{-},{\;H}^{•},{\;\mathrm{OH}}^{•},\;\mathrm{etc}.$$

Hydrogen radicals are strong reducing agents with E_0_ (H^+^/H) = −2.87 V [62]. Solvated electrons and hydrogen radicals are potent reducing agents. Therefore, the silver ions in the solution were quickly reduced to Ag^0^.
(3)$${\mathrm{Ag}}^{+}\;+{\;e}_{\mathrm{aq}}^{-}\to {\;\mathrm{Ag}\;}^{0}$$
(4)$${\mathrm{Ag}}^{+}\;+{\;H}^{•}\;\to {\mathrm{Ag}\;}^{0}+{\;H}^{+}$$

Most of the Ag^+^ ions are consumed when the size of the silver particles reaches a few nanometers. This stops the growth of the particles. In this process, starch serves as both a stabilizer and a template that prevents aggregation of Ag nanoparticles into clusters or blocks [63,64].

### 3.2. Fabrication of AgNPs-PVP Electrospun Fibers

#### 3.2.1. Fibers Morphologies

The SEM images in Figure 6a–d show the electrospun fiber mat products of the 10 wt.% PVP concentration of polymer solution with and without ultrasonic effect. The morphologies of the individual electrospun fibers were smooth, dried, bead-free, and had no useful bonding. Moreover, the surfaces of the electrospun fiber products in Figure 6c,d have a slightly grainy texture, probably due to the addition of starch.

Figure 6c shows that 20 to 50 nm particles were found on the fiber surface. Figure 6c (TEM image) shows that the protrusions on the fiber surface could be agglomerated AgNPs of secondary particle size. Starch could not be dissolved in the polymer-solvent ethanol, and physical stirring could not completely disperse the particles on the micron scale. This resulted in uneven dispersion of AgNPs.

Moreover, no protrusions were observed on the fiber surface after the ultrasonic process, as shown in Figure 6d. Additionally, the mean average diameter of 50 different fibers using the ImageJ software was 0.63 μm. The SEM results indicated that short-term ultrasonic waves in the pre-nozzle area tube had no significant impact on the electrospun fiber mat formation. The results show that US-ES successfully fabricated morphologically suitable electrospun fibers.

#### 3.2.2. Dispersion of AgNPs at Different Concentrations on Electrospun Fibers

Starch-capped AgNPs and PVP ethanol solvents are not miscible. At 0.2, 0.4, and 0.6 wt.% concentration of AgNPs in the PVP polymer solution striped, unevenly dispersed AgNPs were observed on the PVP electrospun fibers, as shown in Figure 7a–c. The dispersion of long strips of nanoparticles could be attributed to the solid chains of starch not being well dispersed in the PVP solution. This could lead to the separation of starch-capped AgNPs and PVP fibers in the electrospinning process. Evaluating the functionality of the unevenly distributed AgNPs in the electrospun fiber products is difficult in practical applications.

In contrast, the AgNPs shown in Figure 7d–f are evenly dispersed on electrospun fibers, with a primary particle mean average size of 1.8–2.0 nm, calculated by measuring 200 different particles using the GMS software. More than 85% of the starch-capped AgNPs solutions had a Ag nanoparticle size distribution between 1 and 2 nm. The randomly agglomerated freeze-dried starch nanoparticles were decomposed and thoroughly mixed with the PVP solution in the pre-nozzle tube before the electrospinning jet due to the ultrasonic effect. The PVP electrospun fibers with 0.6 wt.% starch-capped AgNPs were selected and characterized for the final products in our research.

#### 3.2.3. Analysis of AgNPs-PVP Electrospun Fibers

As shown in Figure 8a, the identity of the characteristic peak for silver (3 keV) was confirmed on the electrospun fiber using EDS analysis. As shown in the STEM image in Figure 8b, highly dispersed AgNPs were observed on the surface of the electrospun fibers. Hence, ultrasonic waves could be an effective method for evenly spreading nanoparticles. Moreover, the solidified PVP could prevent further growth or aggregation of the AgNPs. From Figure 8c, the HRTEM image shows that the AgNPs were uniformly dispersed, and the shape of the nanoparticles was smooth without any edges or corners. Furthermore, particles of the 0.23 nm lattice lines representing silver nanoparticles were observed on the electrospun fiber product. As shown in Figure 8d, the thermal decomposition behaviors of the AgNPs and electrospun fiber samples were characterized using TG-DTA under a nitrogen flow rate of 50 mL/min for a temperature increase of 10 °C/min from 40 to 800 °C. PVP electrospun mats exhibit weight loss between 40 and 100 °C due to moisture evaporation. The primary weight loss between 380 and 460 °C is due to the normal thermal decomposition of PVP. The final weight loss above 470 °C is due to the elimination of carbon residues and silver particles produced by the thermal reaction [27]. Figure 8e shows PVP electrospun mats of 0.6 wt.% starch-capped AgNPs burned using TG-DTA with nitrogen. XPS was performed at a collecting angle of 45° from the average to detect the remaining powder. The peak binding energies of 368.25 eV and 374.35 eV were observed at Ag 3d 5/2 and Ag 3d 3/2, respectively. The XPS results demonstrated that Ag particles were successfully embedded into the PVP electrospun mats [65].

Curve a from Figure 9 shows the FT-IR spectra of pure starch, with the typical absorption bands at 3283 cm^−1^ for OH stretching and 987 and 927 cm^−1^ for COH bending [66]. From curve b, we observed that the starch-capped AgNPs exhibited a curve similar to the curve of pure starch. The signals associated with the OH functional group exhibited a frequency shift at 3296 cm^−1^ of OH stretching as well as at 992 and 930 cm^−1^ of COH bending, confirming the interaction of OH groups with AgNPs [67]. The peaks of silver nanoparticles covered by starch are wider and longer compared to the peaks of pure starch due to intermolecular and intramolecular hydrogen bonds. The existence of the starch characteristic bands in the generated nanoparticles demonstrate that the OH functional groups of the starch-capping agents are effectively bonded with AgNPs [68]. As shown in curves c and d, the two curves exhibit no new peaks, indicating that no new chemical bonds occurred in the PVP electrospun fiber after adding 0.6 wt.% starch to the polymer solution. As shown in curves d and e, a strong absorption peak was observed near 1650 cm^−1^ in both curves. This is a typical feature of the C=O group. The peak in the PVP electrospun fiber occurred at 1654 cm^−1^, whereas the peak in the 0.6 wt.% starch-capped AgNPs-PVP fiber shifted to 1656 cm^−1^, indicating that the Ag ion had some interaction with the C=O group. Moreover, the peak of the C–N bond shifted marginally from 1286 cm^−1^ to 1288 cm^−1^ due to the formation of coordination bonds between the silver and nitrogen atoms generated by the PVP unit [69,70]. The addition of AgNPs resulted in a slight deviation in the peak position of the infrared spectrum. No noticeable shift was observed in the chemical structure of the AgNPs-PVP electrospun fibers.

Curve d is similar to curve e. No apparent shift in peak position or peak intensity changes were observed. Therefore, ultrasonic treatment of US-ES does not alter the chemical structure of the AgNPs-PVP electrospun fibers. This demonstrates the successful fabrication of PVP electrospun fibers containing AgNPs.

#### 3.2.4. Mechanism of Nanoparticles Dispersion by US-ES

As shown in Figure 10a, thousands of cavitation bubbles in liquid form expand, pulse, and collapse under the impact of ultrasonic vibrations. During the cavitation cycle, the microbubbles implosively decomposed, resulting in brief micro hot spots with temperatures above 5000 °C and pressures of approximately 1000 atm [71,72,73]. A transient cavitation bubble can produce an implosion shock. Moreover, this energy is strong enough to break the aggregated starch particles [74,75]. Therefore, the secondary agglomerated starch-capped AgNPs from freeze-drying of approximately 20 nm decompose to form uniform AgNPs of primary size (approximately 2 nm) when the cavitation bubbles explode.

As shown in Figure 10b,c, the strip-shaped nanoparticles were evenly dispersed in the PEEK-insulated tube. The PEEK-insulated tube was tightly wrapped around the ultrasonic generator, making the transmission of the ultrasonic wave highly effective. Moreover, the ultrasonic wave reduced the viscosity of the polymer solution [76], and the acoustic streaming of the ultrasonic wave generated a sonic pressure gradient. The pressure gradient produced microturbulence and liquid jets in the polymer solution, resulting in a potent stirring effect [77,78,79]. Thus, US-ES uniformly dispersed silver nanoparticles in polymer solutions immediately after the electrospinning jet.

## 4. Conclusions

Our research fabricated PVP electrospun fiber mat products with highly dispersed AgNPs. We demonstrated that APDP provides a continuous reaction field in a gas/liquid plasma environment. An amount of 6 mmol/L AgNO_3_ containing 0.1 wt.% starch was used as the feed solution to obtain uniform silver nanoparticles with a mean average size of approximately 2.17 nm. In the process of fabricating 10 wt.% (of polymer solution) PVP electrospun fibers with 0.6 wt.% (of polymer solution) starch-capped AgNPs, US-ES provides an opportunity to fabricate average 0.63 μm PVP fibers, loading a highly dispersed AgNPs with an average size of 2 nm. Finaly, it could be said that this work may provide and update information for the feasibility of developing ultrasonic-based electrospinning for the high dispersion of nanoparticles on electrospun fibers. Utilizing coiled PEEK tube coiling to the ultrasonic generator, which is relatively installable in structure, has the potential to be applied to a portable medical electrospinning device for controlled nanofiber drug release.

## Figures and Tables

**Figure 1 polymers-14-00599-f001:**
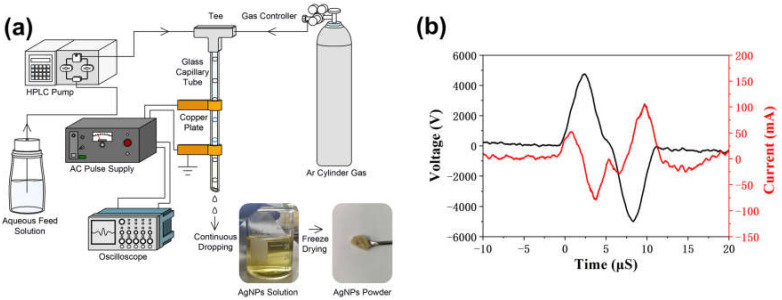
(**a**) Apparatus scheme of synthesizing Ag nanoparticles by atmospheric-pressure pulsed discharge plasma; (**b**) Voltage (black) and Current (red) discharge waveforms.

**Figure 2 polymers-14-00599-f002:**
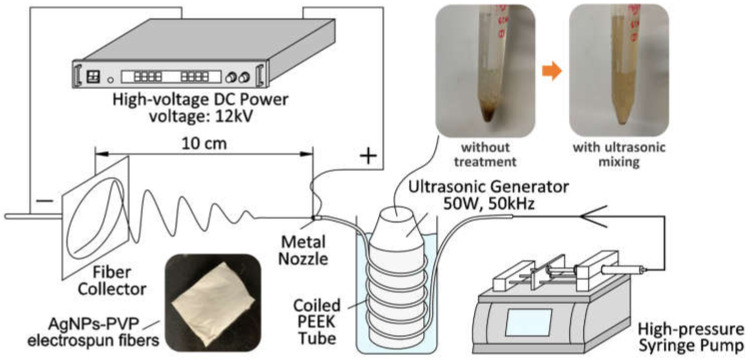
Apparatus scheme of fabricating AgNPs-PVP nanofiber by UN-ES.

**Figure 3 polymers-14-00599-f003:**
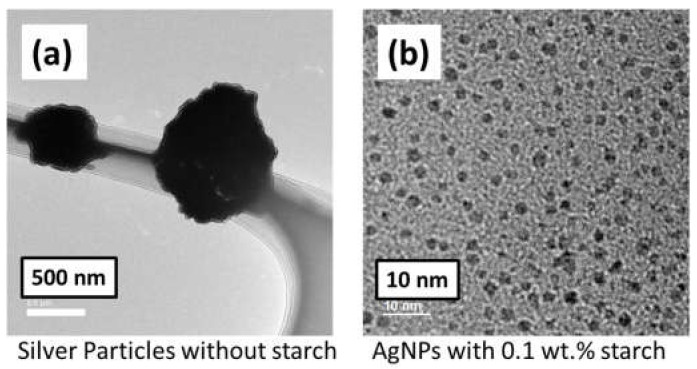
TEM images of (**a**) silver particles without a stabilizer starch. (**b**) silver particles with 0.1 wt.% stabilizer starch.

**Figure 4 polymers-14-00599-f004:**
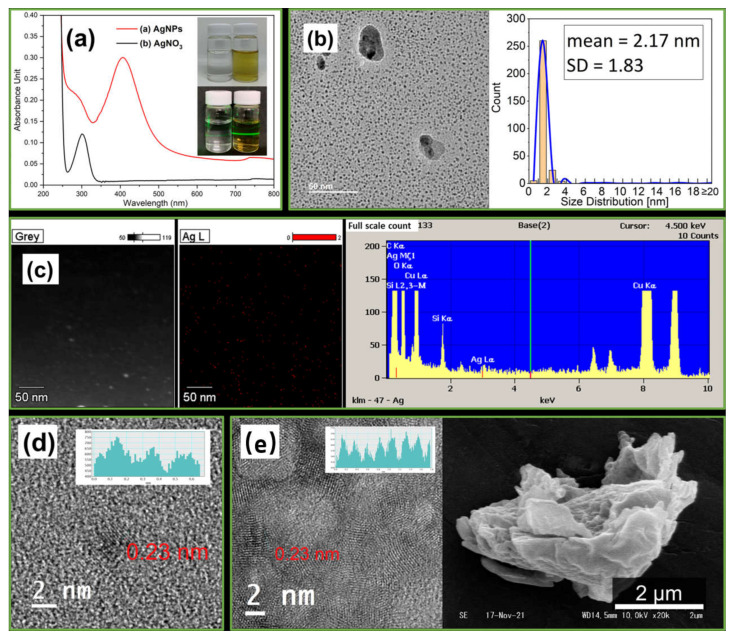
(**a**) UV-Vis and Tyndall Effect photograph of AgNO_3_ feed solution and AgNPs product solution. (**b**) TEM images and size dispersion of AgNPs. (**c**) EDS mapping and spectrum of AgNPs. (**d**) HRTEM lattice lines of the primary particles of AgNPs. (**e**) HRTEM Lattice lines and SEM images of freeze-dried agglomerated secondary particles of starch-capped AgNPs.

**Figure 5 polymers-14-00599-f005:**
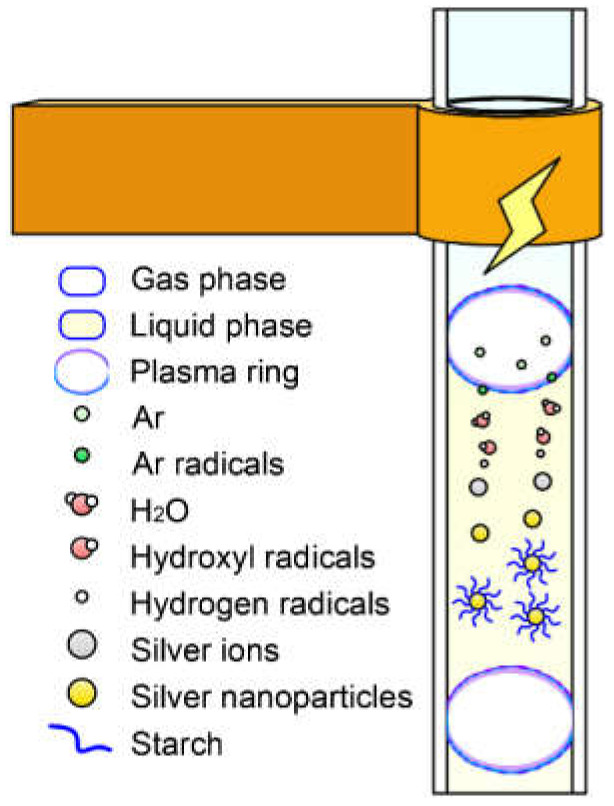
Synthesis mechanism of AgNPs by APDP.

**Figure 6 polymers-14-00599-f006:**
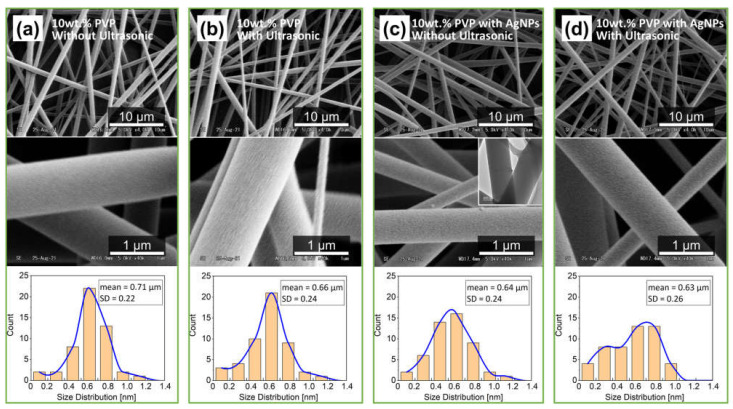
SEM Images of 10 wt.% PVP Electrospun Fibers (**a**) with, (**b**) without Ultrasonic-Assisted. 10wt.% TTO-PVP electrospun fibers (**c**) with, (**d**) without Ultrasonic-Assisted.

**Figure 7 polymers-14-00599-f007:**
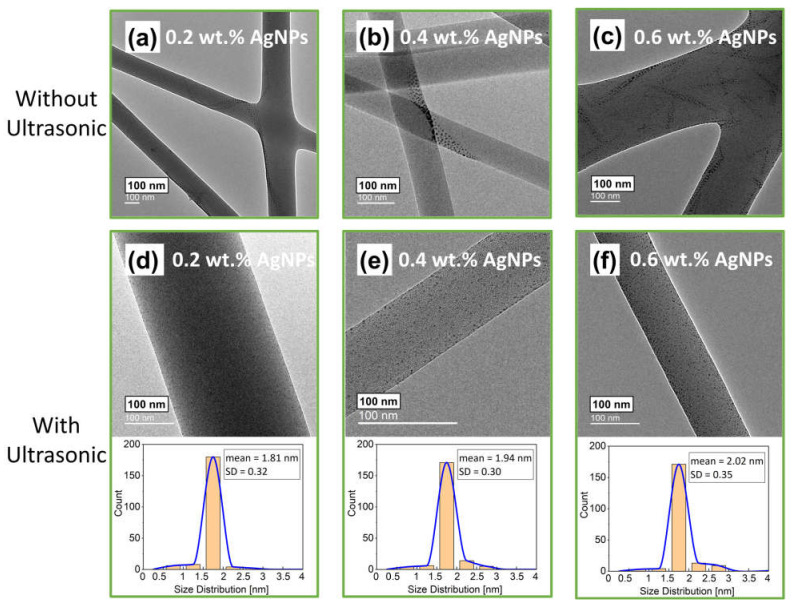
TEM images of (**a**) 0.2 wt.%, (**b**) 0.4 wt.%, (**c**) 0.6 wt.% AgNPs loading on 10 wt.% PVP electrospun fibers without Ultrasonic-Assisted. TEM images and Size Distribution of (**d**) 0.2 wt.%, (**e**) 0.4 wt.%, (**f**) 0.6 wt.% AgNPs loading on 10 wt.% PVP electrospun fibers with Ultrasonic-Assisted.

**Figure 8 polymers-14-00599-f008:**
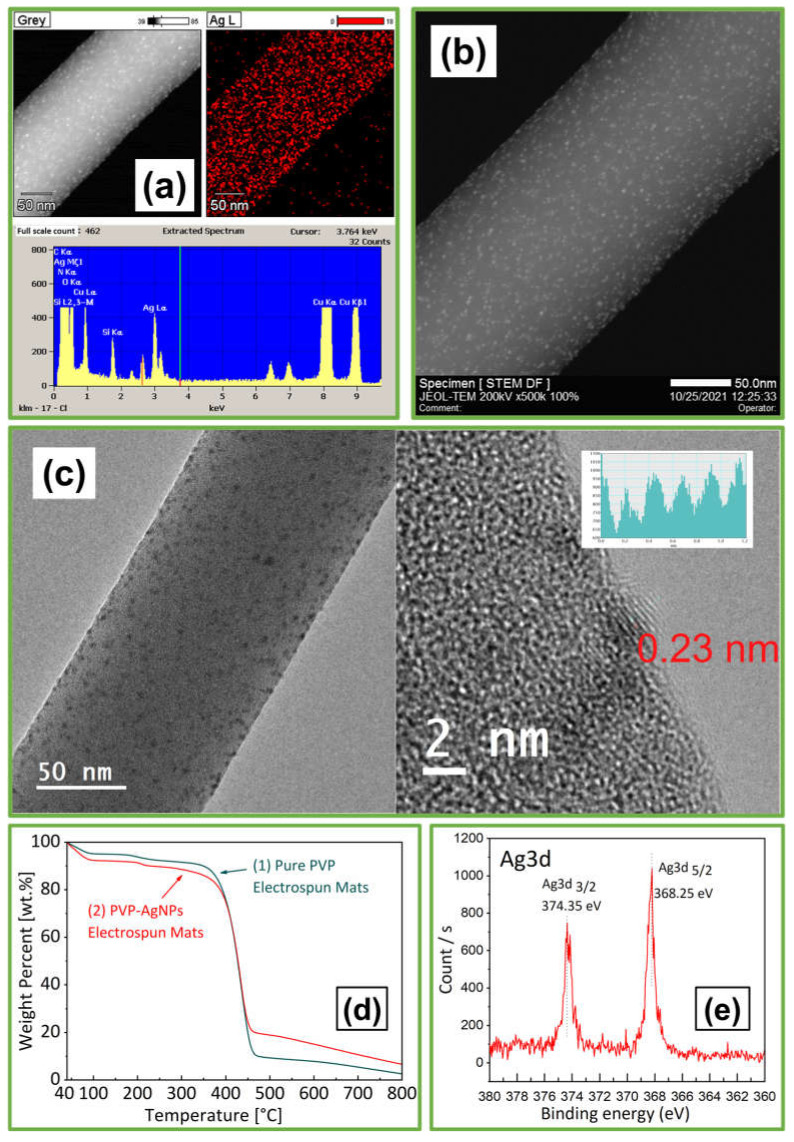
Characterization of 10 wt.% PVP electrospun fibers with 0.6 wt.% AgNPs fabricated by US-ES, including (**a**) EDS mapping and spectrum, (**b**) STEM image, (**c**) HRTEM image, and Lattice lines, (**d**) TG-DTA, (**e**) XPS Analysis.

**Figure 9 polymers-14-00599-f009:**
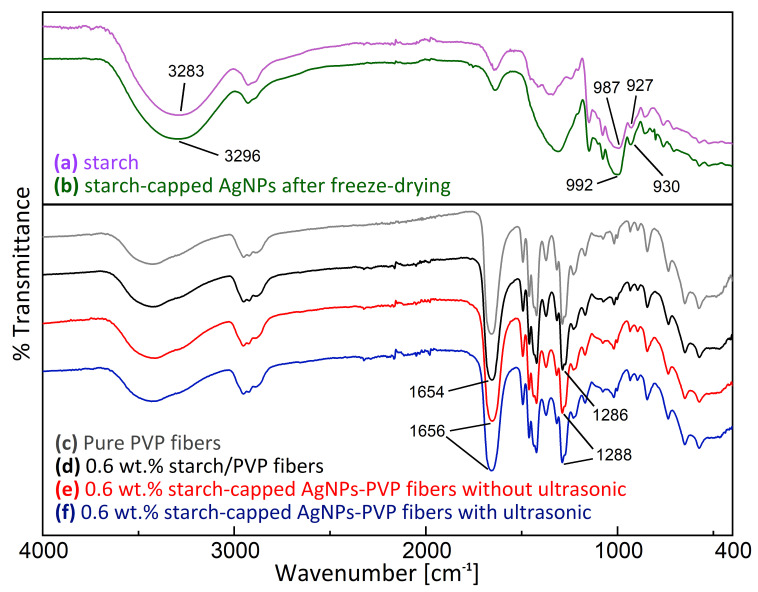
FT-IR Spectra of starch, AgNPs, and PVP fabricated by US-ES.

**Figure 10 polymers-14-00599-f010:**
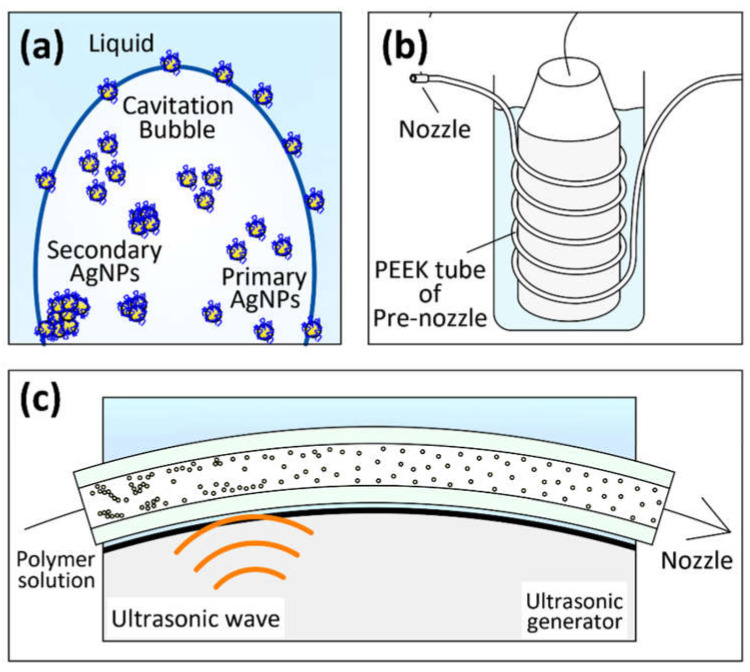
(**a**) Decomposing of agglomerated AgNPs by cavitation bubble; (**b**,**c**) Dispersion of AgNPs in PEEK tube of pre-nozzle area.

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
