# Peer review of "PVP/Highly Dispersed AgNPs Nanofibers Using Ultrasonic-Assisted Electrospinning"

_polymers, 2022, doi:10.3390/polym14030599_

Round 1
Reviewer 1 Report
The manuscript entitled “PVP Electrospun Fiber fabrication with Evenly Dispersed Silver Nanoparticles using Ultrasonic Pre-Nozzle Electrospinning” contains some interesting findings, and it may ultimately be suitable for publication in Polymers. The authors prepared AgNPs-loaded PVP nanofibers and characterized them by using several techniques. There are some issues with the manuscript as it stands (detailed below) and these need to be addressed before it can be considered further. I thus recommend the paper be reconsidered after major revisions.
The reviewer has the following comments
- The title of the manuscript is confusing, it should be modified.
- Rewrite the abstract. For instance, modify the following text
“Electrospun fibers containing silver nanoparticles (AgNPs) are used in antibacterial materials, medical device coatings, optical sensors, cosmetics, electronic composite materials, and food packaging. We fabricated polymer fibers with uniformly dispersed AgNPs by electrospinning a polymer solution mixed with AgNPs using ultrasonic treatment”.
- The statistical analysis section should be added to the revised manuscript.
- What are the major applications of AgNPs loaded PVP fibers? Are these nanofibers biocompatible?
- The introduction section is very short and should be revised entirely so that the reader can clearly identify the scientific problems solved by this research. There are several systems with a wide range of biomaterials, including hydrogels, nanoparticles, nanofilms, nanocomposites, etc., but why did the authors select only nanofibers? The authors should emphasize why the nanofibers are familiar, or favor compared to other systems using the following articles. Moreover, the information on biomaterials should be elaborated in the introduction with recent references (preferably 2020-2021), Thus, the following articles should be quoted in the introduction.
Chitosan (https://doi.org/10.1016/j.colsurfb.2021.111819),
Gelatin (https://doi.org/10.3390/ph14040291,https://doi.org/10.1016/j.jmbbm.2020.103696) PCL (https://doi.org/10.1016/j.msec.2020.110928) and others
https://doi.org/10.1039/D1EN00354B, https://doi.org/10.1016/j.msec.2020.111698, https://doi.org/10.1016/j.msec.2020.110928
It would be more realistic to cover such kind of research work in the current manuscript. Which will enrich the quality of the current manuscript as well as inquisitiveness to the readers.
- According to the revised data, the inclusions should be modified with more quantitative data.
Author Response
Thank you very much for your questions and suggestions for improving the manuscript!
1. The title of the manuscript is confusing, it should be modified.
The manuscript title we rewrote is as follow: “PVP/ Highly Dispersed AgNPs Nanofibers using Ultrasonic-assisted Electrospinning”
2. Rewrite the abstract. For instance, modify the following text
“Electrospun fibers containing silver nanoparticles (AgNPs) are used in antibacterial materials, medical device coatings, optical sensors, cosmetics, electronic composite materials, and food packaging. We fabricated polymer fibers with uniformly dispersed AgNPs by electrospinning a polymer solution mixed with AgNPs using ultrasonic treatment”.
Thank you very much. The manuscript abstract we rewrote is as follow:
“Silver nanoparticles (AgNPs) are novel materials with antibacterial, antifungal, and antiviral activities over a wide range. This study aimed to prepare polyvinylpyrrolidone (PVP) electrospinning composites with uniformly distributed AgNPs. In this study, starch-capped ~2 nm primary AgNPs were first synthesized using Atmospheric pressure Pulsed Discharge Plasma (APDP) at AC 10 kV and 10 kHz. Then, 0.6 wt.% AgNPs were mixed into a 10 wt.% PVP ethanol-based polymer solution and coiled through Ultrasonic-assisted Electrospinning device (US-ES) with the 50 W and 50 kHz ultrasonic generator. At 12 kV and a distance of 10 cm, this work successfully fabricated AgNPs-PVP electrospun fibers. Electrospun products were characterized using Scanning Electron Microscopy (SEM), Transmission Electron Microscopy (TEM), High-Resolution TEM (HR-TEM), Fourier Transform Infrared Spectroscopy (FT-IR), X-ray Diffraction (XRD), Thermogravimetric (TG), and X-ray Photoelectron Spectroscopy (XPS) methods.”
3. The statistical analysis section should be added to the revised manuscript.
We took this suggestion to be for additional statistical data on size distribution. Rather than adding a section on statistical processing, we decided it would be easier to incorporate the data into the figures, and made the following revisions. In Figure 4(a), Figure 6 and Figure 7, we corrected the column of the particle size distribution and added standard deviation calculations of the statistical analysis.
4. What are the major applications of AgNPs loaded PVP fibers? Are these nanofibers biocompatible?
We added the introduction for the application of the fibers and references of nanofibers biocompatible as follow:
The applications of AgNPs loaded PVP fibers we added around the end of the Introduction as follow :
“AgNPs loaded PVP fibers could be applied as glucose sensors [57], and biocompatible materials include antimicrobial mats [58] and wound dressing materials [59]. ”
We also added some citations to describe the nanofiber is biocompatible:
“It can be effectively used in medicine, pharmacy, cosmetics, and composite material applications due to its non-toxic, biocompatibility, temperature resistance, pH stability, and chemical inertness [53-55].”
5. The introduction section is very short and should be revised entirely so that the reader can clearly identify the scientific problems solved by this research. There are several systems with a wide range of biomaterials, including hydrogels, nanoparticles, nanofilms, nanocomposites, etc., but why did the authors select only nanofibers? The authors should emphasize why the nanofibers are familiar, or favor compared to other systems using the following articles. Moreover, the information on biomaterials should be elaborated in the introduction with recent references (preferably 2020-2021), Thus, the following articles should be quoted in the introduction.
Chitosan (https://doi.org/10.1016/j.colsurfb.2021.111819),
Gelatin (https://doi.org/10.3390/ph14040291,
https://doi.org/10.1016/j.jmbbm.2020.103696)
PCL (https://doi.org/10.1016/j.msec.2020.110928)
and others
https://doi.org/10.1039/D1EN00354B,
https://doi.org/10.1016/j.msec.2020.111698,
https://doi.org/10.1016/j.msec.2020.110928
It would be more realistic to cover such kind of research work in the current manuscript. Which will enrich the quality of the current manuscript as well as inquisitiveness to the readers.
Thank you very much. The introduction of this manuscript was rewritten, and also it added more experimental background and recent references as follow:
“Here, starch has been used as a green capping agent for AgNPs [10]. ”
“It is desirable to attach nanoparticles to composite materials, which can be applied in applications including drugs encapsulation and controlled release [11-13], food preservation [14], wastewater treatment [15], gas sensing [16], and environmental remediation [17,18]. In this work, as one of the composite materials, nanofiber was selected due to its high surface area to volume ratio, efficient mechanical properties, and specific biochemical properties [19-22]. Here, the electrospinning technique is an efficient method of fabricating nanofibers [23,24].”
“It was well known that ultrasonic is a commonly used method for the deagglomeration and dispersion of nanoparticles into liquids [36-39]. Some studies added surfactants [40,41] and stabilizers [42] simultaneously as ultrasonic treatment.”
6. According to the revised data, the inclusions should be modified with more quantitative data.
Additionally, we added measurements of the diameter of electrospun fibers of section 3.2.1 Fibers morphologies.

Reviewer 2 Report
The present manuscript describes the preparation of PVP nanofibers containing well-dispersed Ag NPs by US-ES. Before publication, authors must address the following:
-More physicochemical data of the AgNPs should be provided (surface charge, FTIR, EDS). The TEM image still shows 4 agreggates.
-Real images of the apparatus used should be reported (for both atmospheric-pressure pulsed discharge plasma and US-ES).
-Authors should comment if the US-ES approach would work for well-dispersed AgNPs attained by techniques other than that herein reported.
-Improve the discussion. Authors should compare their findings and their technique with others reported elsewhere. Future directions must be provided.
Author Response
Thank you very much for your questions and suggestions for improving the manuscript.
1. More physicochemical data of the AgNPs should be provided (surface charge, FTIR, EDS). The TEM image still shows 4 aggregates.
Thank you very much for your kind suggestions. We have made the following revisions to FTIR and EDS as you indicated.
For the FTIR of AgNPs: Please see Figure 9 curve(b) of the FT-IR of Starch-capped AgNPs.
For the EDS of AgNPs: Please see Figure 4 (c) of EDS mapping and spectrum of AgNPs and related text explanation.
Regretfully, due to the current limitations of our analytical equipment, we are not able to identify the surface charge of AgNPs. Already AgNPs solutions were directly freeze-dried in this work. We decided to explain the aggregation state as an alternative knowledge to surface charging with this point.
The manuscript adds the following text explanation for the 4 aggregates of the TEM image of Figure 4 (b):
“Nevertheless, several agglomerates on the TEM support films were still found. Probably, it came from the aggregation of starch-capped primary-sized particles to form secondary-sized particles during the AgNPs solution droplets dried. We calculated the mean average size (2.17 nm) and standard deviation (SD = 1.83) of the particles by measuring the sizes of 300 randomly selected particles using the Gatan Microscopy Suite (GMS) software.”
“Figure 4(c) of the EDS mapping and spectrum shows that the silver element was detected in the starch-capped AgNPs product.”
2. Real images of the apparatus used should be reported (for both atmospheric-pressure pulsed discharge plasma and US-ES).
[Please check the word attachment document for related images, thank you very much! ]
Above are the real images of the atmospheric-pressure pulsed discharge plasma and US-ES apparatus. As can be seen, it is not easy to understand the connection of the device using real images. To make it easier for the reader to understand, we have created schematic diagrams in the original manuscript, as shown in Figure 1 and Figure 2.
3. Authors should comment if the US-ES approach would work for well-dispersed AgNPs attained by techniques other than that herein reported.
Thank you very much. We rewrote the introduction section of this manuscript. We illustrate the use of ultrasound as a commonly used nanoparticle dispersion technique and the use of US-ES equipment to produce highly dispersed AgNPs attained fibers without adding surfactants, as follows:
“Mechanical stirring for enhancing dispersion uniformity [34] cannot prevent particles from aggregating [35]. It was well known that ultrasonic is a commonly used method for the deagglomeration and dispersion of nanoparticles into liquids [36-39]. Some studies added surfactants [40,41] and stabilizers [42] simultaneously as ultrasonic treatment.”
“This study finally fabricated the PVP electrospun fibers with highly dispersed AgNPs using APDP and US-ES without adding surfactants. ”
4. Improve the discussion. Authors should compare their findings and their technique with others reported elsewhere. Future directions must be provided.
We have made revisions as possible according to the reviewer's suggestions. Also, we have added the prospect of AgNPs loaded PVP fibers in the introduction as follow:
“AgNPs loaded PVP fibers could be applied as glucose sensors [57], and biocompatible materials include antimicrobial mats [58] and wound dressing materials [59]. ”
In the Conclusion section, we added the following future directions:
“Eventually, it could be said that this work may provide and update information for the feasibility of developing ultrasonic-based electrospinning for the high dispersion of nanoparticles on electrospun fibers. Utilizing coiled PEEK tube coiling to the ultrasonic generator, which is relatively installable in structure, has the potential to be applied to a portable medical electrospinning device for controlled nanofiber drug release.”

Round 2
Reviewer 1 Report
The authors have clarified all my concerns, and the quality of the manuscript was significantly improved. I must congratulate the authors for their willingness in addressing the reviewer’s comments. So, I recommend accepting the manuscript in its present form.